

# Direct production of fermionic superfluids
# in a cavity-enhanced optical dipole trap

Tabea N. C. Bühler, Timo Zwettler, Gaia S. Bolognini, Aurélien H. Fabre,
Victor Y. Helson°, Giulia Del Pace⋆† and Jean-Philippe Brantut

Institute of Physics and Center for Quantum Science and Engineering,
École Polytechnique Fédérale de Lausanne, CH-1015 Lausanne, Switzerland

⋆ delpace@lens.unifi.it

## Abstract

We present the production of quantum degenerate, superfluid gases of $^6$Li through direct evaporative cooling in a cavity-enhanced optical dipole trap. The entire evaporative cooling process is performed in a trap created by the TEM$_{00}$ mode of a Fabry-Pérot cavity, simultaneously driven on several successive longitudinal modes. This leads to near-complete cancellation of the inherent lattice structure along the axial direction of the cavity, as evidenced by the observation of long-lived dipole oscillations of the atomic cloud. We demonstrate the production of molecular Bose-Einstein condensates upon adiabatic conversion of a unitary Fermi gas evaporatively cooled in this trap. The lifetime and heating in the cavity trap are similar to those in a running wave dipole trap. Our system enables the optical production of ultracold samples using a total trap-laser power below $1$ W, leveraging the benefits of optical resonators as dipole traps in quantum gas research while maintaining a simple resonator design and minimizing additional experimental complexity.

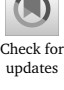

## Contents

---

° Current affiliation: Swiss Center for Electronics and Microtechnology (CSEM), CH-2000 Neuchâtel, Switzerland.
† Current affiliation: University of Florence, Physics Department, Via Sansone 1, 50019 Sesto Fiorentino, Italy.

# 1   Introduction

All protocols to produce quantum degenerate gases rely on magnetic or optical forces to trap and cool down atoms [1, 2]. In optical dipole traps, far-detuned light from the main atomic resonances is used to create quasi-harmonic traps at the focus of a laser beam, where atoms are trapped at the energy minimum. For these traps, large detunings are necessary to limit photon scattering, requiring high optical powers to reach the potential depth necessary to capture laser-cooled atoms and to ensure high collision rates [3–5].

Power buildup in optical resonators is an elegant technique to enhance the optical power in dipole traps, offering enhancements by factors up to tens of thousands for the highest finesse [6–13]. In addition, a cavity-based dipole trap benefits from built-in mode cleaning, such that the trap shape is guaranteed to maintain a fixed Gaussian profile for all optical powers and thermal lensing effects are mitigated. This stability is an important asset for evaporative cooling where laser power is varied over several orders of magnitude. However, in the common Fabry-Pérot configuration, the mode has an intrinsic standing-wave structure along the cavity direction, creating an optical lattice. While this does not preclude the use of the cavity for capturing atoms after laser cooling, it hinders evaporative cooling of the gas into a quantum degenerate many-body state due to the inability of atoms in separate wells of the lattice to collide with each other.

This issue can be solved by using a ring resonator with traveling wave modes [9, 10, 13], at the cost of an increased complexity, in particular due to the increased cavity length and correspondingly low linewidth. Another attractive method, that allows to keep the Fabry-Pérot configuration of the cavity, consists in driving the cavity simultaneously with several frequencies, separated by multiples of the free-spectral range. For a particular choice of the relative intensities in the different spectral components, the inherent lattice structure of the cavity is cancelled over a finite range [14]. Enabling spatially homogeneous light-matter coupling while preserving the advantages of a simple cavity design, this scheme is beneficial not only for shaping optical dipole traps [14–18], but also other applications, including the generation of homogeneous cavity probe fields [19, 20]. However, for the purpose of producing quantum degenerate gases through evaporative cooling, additional dipole traps in free space have been necessary so far.

Here, we reliably produce fermionic superfluids by direct evaporation in a Fabry-Pérot cavity trap, that serves both for capturing the laser-cooled atoms and, by means of the lattice-cancellation scheme [14], to create the final optical dipole trap. The paper is structured as follows. In Sec. 2, we describe the used lattice-cancellation scheme, its experimental implementation and the cooling procedure. We further describe how we optimize the lattice cancellation and the optical evaporation. In Sec. 3 we confirm successful evaporative cooling of the atomic cloud into the superfluid regime inside the lattice-cancelled cavity trap by observing the onset of a finite condensate fraction. We further present the performance of the cavity trap in the final configuration, quantifying the lifetime and heating. Finally, in Sec. 4 we discuss the conclusions and perspectives of our work.

# 2   Lattice-cancelled cavity trap and cooling procedure

## 2.1   Lattice-cancellation scheme

The longitudinal modes of a Fabry-Pérot optical cavity exhibit a standing-wave structure, leading to a one-dimensional lattice potential when used to trap atoms. However, this lattice can be efficiently suppressed by using several longitudinal modes simultaneously. When the fre-

quency difference between these modes is large compared to other relevant frequencies in the system, the atoms experience a potential given by the sum of the individual mode contributions. In a symmetric cavity, successive modes are dephased by $\pi/2$ at the cavity center, such that the total potential resulting from two identically driven consecutive modes is nearly uniform over a finite region close to the cavity center.

In our experiment (see [21] for details), we use a cavity with a finesse $\mathcal{F} = 3.6(1) \cdot 10^3$ at 1064 nm, offering a buildup larger than $10^3$ and a free spectral range of $\omega_{\text{FSR}} = 2\pi \times 3.6277(1)$ GHz. The beam waist at the cavity center is 56.6(3) μm inferred from transverse mode spectroscopy. To inject light simultaneously on several consecutive longitudinal modes, we drive the cavity with a phase-modulated laser beam with a modulation depth $\beta$, such that its spectrum features sidebands separated in frequency by multiples of $\omega_{\text{FSR}}$. To introduce the modulation, we use a fibered electro-optical modulator[1] (EOM) in combination with a tunable gain amplifier.[2] The beam passing through the phase modulator is referred to as CODT2 in the following of this paper. Its spectrum, after having passed the EOM, is depicted in Fig. 1a. The intensity of a given sideband $n$ produced by the EOM for a modulation depth $\beta$ is described by the Bessel function of first kind $J_n(\beta)$. The contributions to the optical potential $V$ inside the cavity of the carrier $J_0$ and the first two sidebands $J_1$, $J_2$ are shown in Fig. 1b, as well as the expected total potential for a modulation depth of $\beta = 1.20$ rad. This optimal value is obtained by minimizing the peak to peak lattice of the total optical potential along the cavity axis at its center. It leads to an expected residual lattice $\Delta V$ of less than 1% over a length scale of 2 mm, which is one order of magnitude larger than the typical extent of our atomic cloud. This is illustrated in Fig. 1c.

This prediction of the optimal modulation depth does not account for potential complications that may arise in practice, such as variations in the free spectral range of the cavity due to the frequency dependence of the characteristics of the mirror coating. In our experiment, the frequency range explored within the scope of the presented scheme is on the order of tens of GHz. Within this range, we did not observe any measurable differences in the free spectral range of the cavity. In cases where such variations are present, the optimal modulation depth required to minimize the lattice structure at the center of the cavity is expected to change. However, as long as the variations in the free spectral range remain small compared to the cavity linewidth, neither the predicted optimal modulation depth nor the quality of the lattice cancellation is expected to change significantly.

In addition to CODT2, we couple an unmodulated beam (CODT1) originating from the same laser source but with orthogonal polarization compared to CODT2 to the optical cavity, as depicted in Fig. 1a. This beam serves in the loading phase from the magneto-optical trap, circumventing power limitations of the fibered EOM. In earlier instances of the experiment, we used a free-space resonant EOM to overcome the power limitation. However, we observed drifts of the modulation depth over hours, making the overall sequence less stable. Both trap laser powers are stabilized by a feedback loop measuring the power injected into the cavity for CODT1 and the power transmitted through the cavity for CODT2. In addition, the cavity length is stabilized throughout the entire duration of the experimental sequence on the first harmonic of the dipole trap light at 532 nm. In total, we employ about 1 W of laser power to produce both CODT1 and CODT2, enough to fulfill our experimental requirements, taking into account the finite efficiency of the acousto-optical modulators and fiber couplings used in the preparation of the two beams.

---

[1]Exail, model: NIR-MPX-LN-02-00-P-P-FA-FA.
[2]Exail, model: DR-AN-10-MO.

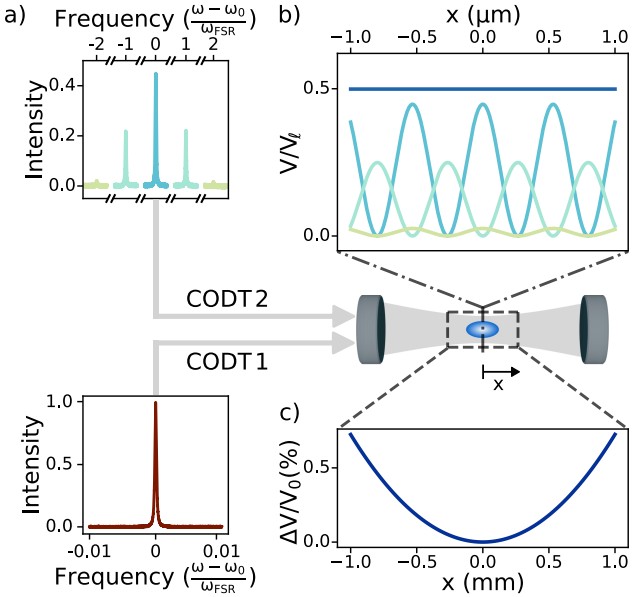

Figure 1: Experimental setup used to perform evaporative cooling to degeneracy inside a Fabry-Pérot optical cavity. a) Two beams (CODT1, CODT2) are injected to the cavity. CODT1 assists in the loading phase from the magneto-optical trap of the experimental sequence. CODT2 exhibits a spectrum consisting of a carrier and several sidebands, separated by multiples of the free spectral range $\omega_{FSR}$ and is used to produce the final, lattice-free trap. b) Total optical potential originating from CODT2 (——) as well as individual contributions of carrier (——), first (——) and second sidebands (——), given relative to the peak lattice potential $V_\ell$ of a single, non-modulated beam of the same strength. c) Residual lattice modulation of the optical potential along the cavity axis $\Delta V$ normalized with the potential depth at the cavity center $V_0$.

## 2.2 Cooling sequence

The evaporative cooling sequence starts with a gas of laser-cooled $^6$Li atoms in a compressed magneto-optical trap operating on the D2 line. The CODT1 is on during the laser cooling process with about 150 mW of cavity input power. In this regime, heating effects do not impose a limitation on the operation of the cavity. Considering the power buildup, this corresponds to a peak lattice depth of 8.3 mK inside the cavity and yields approximately $7.2 \cdot 10^6$ atoms captured in a one-dimensional optical lattice, without the need for advanced laser cooling protocols [22, 23]. The magnetic field is then ramped close to 832 G, the location of a broad Feshbach resonance, where the scattering length between the two lowest hyperfine states of $^6$Li diverges. In this configuration, the magnetic field residual curvature along the cavity direction yields an underlying harmonic trap with a frequency of $\omega_x/2\pi = 28$ Hz. A first linear ramp of CODT1 down to 20 mW cavity input power over 400 ms is performed to start the evaporation. In parallel with the first evaporation ramp, we ensure having a balanced mixture of the two lowest hyperfine states through a Landau-Zener sweep of the radio frequency (RF) field coupling those two states. We found that for similar laser powers, loading directly into the lattice-cancelled trap yields worse performance in terms of final atom numbers, suggesting that the lattice helps maintaining high density at the early stage of the evaporation, preventing the atoms to spread along the cavity direction and ensuring high collision rates.

Subsequently, the atoms are transferred into the lattice-free dipole trap CODT2, after which we perform a linear evaporation ramp with a duration of 1 s down to a cavity input power on

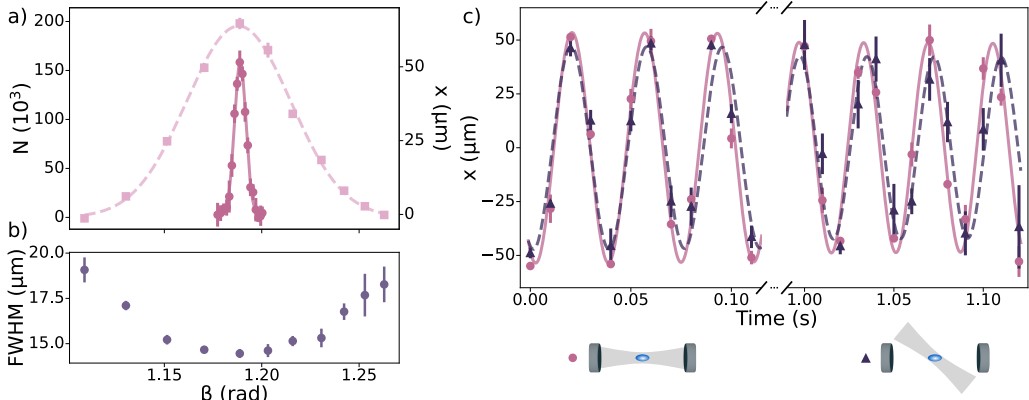

Figure 2: Optimization of the lattice cancellation. a) Atom number per hyperfine state $N$ at a fixed evaporation endpoint (■) and amplitude of dipole oscillations measured as the displacement of the atomic cloud in the x-direction after three oscillation periods (●), both as a function of modulation depth $\beta$ of the CODT2. The dashed (- - -) and solid (——) lines represent Gaussian fits to the data of corresponding color. b) Transverse size of the cloud as a function of modulation depth $\beta$. c) Longitudinal dipole oscillations of the cloud trapped in the cavity trap along the x-direction (●) and, for comparison, of the cloud trapped in a running-wave optical dipole trap (▲) as a function of time. The solid (——) and dashed (- - -) lines represent a fit of an exponentially damped cosine function. All error bars represent the standard deviation of repeated measurements.

the order of $100\,\mu W$, corresponding to a power on the order of hundreds of mW inside the resonator, considering its power buildup. To further decrease the temperature of the cloud we linearly increase a magnetic field gradient along the radial direction with respect to the cavity axis, keeping the optical dipole trap power constant. This effectively reduces the trap depth at constant trapping frequency, improving the evaporation efficiency in the final stage [24]. The magnetic field gradient is removed after the evaporation sequence.

We first optimize the lattice cancellation based on the outcome of the evaporative cooling sequence. To this end, we perform evaporative cooling down to a fixed optical power, corresponding to a measured radial trapping frequency of $\omega_r/2\pi = 430\,Hz$ and a trap depth of $4.5\,\mu K$, for different modulation depths $\beta$ in CODT2. We measure the number of atoms per hyperfine state $N$ and monitor *in situ* the transverse size of the cloud as a measure of the internal energy. The results are presented in Fig. 2a and Fig. 2b, showing a clear optimum with maximal atom number and minimal size. By measuring the sideband amplitude directly using cavity transmission spectroscopy, we deduce the modulation depth realizing the maximal atom number $\beta = 1.19(1)\,rad$, where the uncertainty stems from the cavity spectroscopy. Being close to the theoretical expectation $\beta = 1.20\,rad$ to achieve lattice cancellation, this indicates, as anticipated in previous studies [17], that the key element hindering efficient evaporation in cavity dipole traps is the lattice structure.

## 2.3 Dipole oscillations

We now optimize the lattice cancellation independently of the evaporative cooling sequence by triggering dipole oscillations along the cavity axis and measuring their amplitude after a fixed duration as a proxy for the damping rate. To this end, we shift the location of the magnetic field saddle point along the x-direction using compensation coils and suddenly bring it back to excite dipole oscillations, which we monitor by tracking the *in-situ* cloud position over time. Also here

we find an optimal value of $\beta$, for which the oscillations are presented in Fig. 2c, showing a decay by less than 10% over one second. When deviating from the optimal $\beta$, we observe a sharp decrease in the measured oscillation amplitude, indicating an increase of the damping rate. Fig. 2a shows the measured amplitude of the dipole oscillations after three oscillation periods. The peak is approximately Gaussian, with a fitted full width at half maximum of 0.007 rad, one order of magnitude sharper than the peak of atom number after evaporation but yielding the same optimal location. This demonstrates that the optimal evaporative cooling conditions are without a remaining lattice structure in the optical potential.

Last, to benchmark the observed damping of the dipole oscillations, the same measurement is performed using atoms trapped in a running wave dipole trap. To do so, at the end of the evaporation procedure, we transfer the atoms into a dipole trap created by a running-wave, focused laser beam intersecting the cavity axis at an angle of 18°, and we switch off the CODT2. Using the same procedure, we measure dipole oscillations in this trap as shown in Fig. 2c. We observe a similar damping as for the cavity dipole trap, confirming that the cavity-assisted trap provides a harmonic confinement and does not represent any compromise in terms of potential shape compared with dipole traps used in previous experiments with ultracold gases.

# 3 Fermionic superfluids in the cavity trap

We now demonstrate the production of superfluid Fermi gases through evaporation in the lattice-free optical dipole trap. As a two-component attractive Fermi gas is cooled, it undergoes a phase transition to the superfluid state at a critical temperature $T_C$. For positive values of the interaction parameter $1/k_F a$, a convenient manifestation of superfluidity is a finite condensate fraction $N_0/N$ observed in the density profile of the atomic gas after time of flight, indicating Bose-Einstein condensation of molecules [25].

Here, we perform evaporative cooling of the Fermi gas close to unitarity and subsequently ramp the magnetic field to the molecular side of the Feshbach resonance. There we measure $N_0/N$, whose variations are equivalent to temperature changes. Observing a finite condensate fraction indicates that the molecular gas is in the superfluid phase after the adiabatic ramp, indirectly proving that the unitary gas was superfluid prior to the ramp [26]. We chose to monitor $N_0/N$ because it has proven to be a robust indicator across a wide range of parameters for our purposes.

Fig. 3a shows $N_0/N$ on the molecular side of the Feshbach resonance at 747 G, corresponding to $1/k_F a = 1.2$, as a function of cavity input power $P$ at the endpoint of optical evaporation performed close to unitarity. $N_0/N$ is extracted from a fit of a bimodal density profile to the integrated time-of-flight image of the cloud after averaging over several individual images. The time-of-flight duration is 8.5 ms. $N$ denotes the atom number per hyperfine state and is measured for each experimental repetition by means of cavity spectroscopy as explained in [17]. The error bars of $N_0/N$ represent the standard deviation of the bootstrap distribution, obtained by repeatedly fitting randomly sampled, averaged images. In Fig. 3a (i)-(iii), three examples of averaged, integrated time-of-flight images are shown. The solid lines are bimodal fits to the integrated column densities, that are performed to extract the condensate fractions. An example of an *in-situ* picture of the atomic cloud at $1/k_F a = 1.2$ with a condensate fraction of $N_0/N = 0.14$ is shown in Fig. 3b. In addition, we extract the temperature of the gas at unitarity for a fixed evaporation endpoint from a fit of the equation of state to the two-dimensional atomic density. For a cloud with $N = 3 \cdot 10^5$ atoms per hyperfine state we extract a temperature of $T/T_F = 0.157(3)$, where $k_B T_F = \hbar \overline{\omega} (6N)^{1/3}$ and $\overline{\omega}$ denotes the geometric mean of the trapping frequencies, which are $\omega_r/2\pi = 430$ Hz and $\omega_x/2\pi = 28$ Hz in the radial and longitudinal direction respectively, at the endpoint of evaporation.

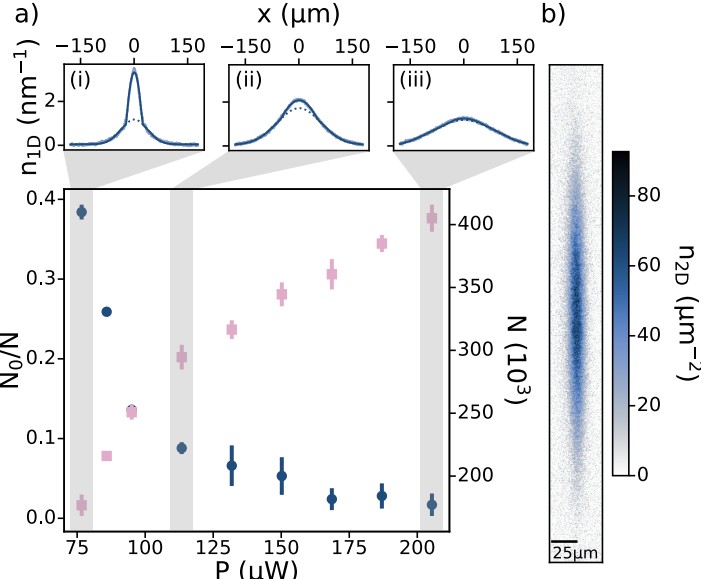

Figure 3: Production of fermionic superfluids in a lattice-cancelled cavity dipole trap.
a) Condensate fraction $N_0/N$ (●) and atom number per hyperfine state $N$ (■) mea-
sured on the molecular side of the Feshbach resonance with an interaction parameter
$1/k_F a = 1.2$, following evaporation close to unitarity and an adiabatic ramp of the
magnetic field. Both quantities are given as a function of cavity input laser power
$P$ at the endpoint of optical evaporation. $N_0/N$ is extracted from a fit of a bimodal
density distribution to the average integrated density profile. Error bars of $N_0/N$
represent the standard deviation of the bootstrap distribution. Error bars of atom
number represent the standard deviation of repeated measurements. (i) - (iii) Three
examples of bimodal fits (——) to the average integrated column density $n_{1D}$ (●).
The dashed lines (----) mark the thermal part obtained by the fit of the bimodal
density profile. b) *In situ* absorption image showing the column density $n_{2D}$ of the
atomic gas with an interaction parameter $1/k_F a = 1.2$.

To fully validate the potential of the cavity-assisted trap for future quantum gas applica-
tions, we evaluate the lifetime and quantify heating of the superfluid gas in the presence of
lattice cancellation. We first measure the atom number $N$ of a superfluid Fermi gas held close
to unitarity in the cavity-assisted trap after the last step of evaporation as a function of hold
time. The results are shown in Fig. 4a, where we restrict the measurement duration to 2.5 s
due to heating of the electromagnets. Extrapolating the observed decay with an exponential fit
yields a characteristic decay time, interpreted as lifetime, of 7.6(3) s, which is on the same or-
der of magnitude as in our previous experiments with a crossed dipole trap [17]. We attribute
the observed lifetime primarily to mechanisms other than spontaneous scattering of dipole
trap photons. These include background collisions, three-body recombination processes and
intensity noise in the dipole trap, independent of the cavity.

We further quantify heating as a function of time in the unitary Fermi gas. After a variable
holding time in the unitary regime we adiabatically ramp the magnetic field to the molecular
side of the Feshbach resonance and measure the condensate fraction $N_0/N$. The results are
shown in in Fig. 4b. We measure a finite condensate fraction up to hold times of several hun-
dreds of ms, compatible with demanding applications requiring long-time observation as for
probing fermionic superfluids [27–29]. The observed time scale associated to heating is similar
to that observed in our previous experiments with a crossed dipole trap [17], demonstrating
that the lattice-cancelled cavity trap maintains the total duration available for experiments.

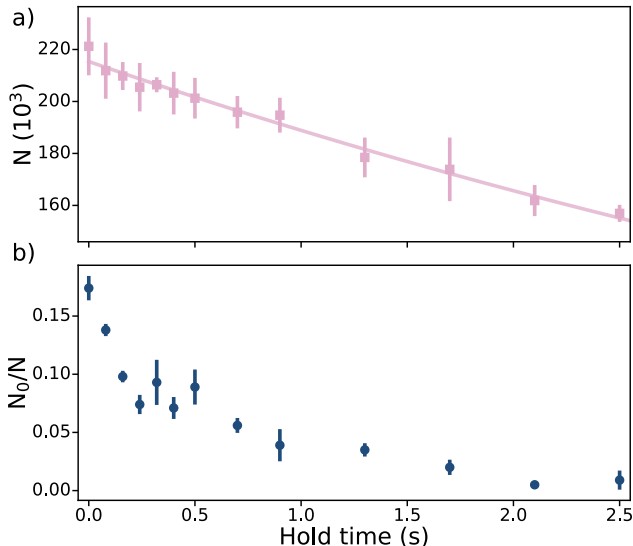

Figure 4: Characteristic decay time and heating in the lattice-free cavity trap. a) Measured atom number per hyperfine state $N$ of a unitary Fermi gas as a function of hold time close to unitarity in the cavity trap. The solid line shows an exponential fit from which we extrapolate a lifetime of 7.6(3) s. Error bars represent the standard deviation of repeated measurements. b) Condensate fraction $N_0/N$ extracted from a bimodal fit to the average integrated density profile as a function of hold time to quantify heating of the atomic gas. During the hold time the magnetic field is kept at 832 G. Subsequently, we adiabatically sweep the magnetic field to the molecular side of the Feshbach resonance at 747 G corresponding to an interaction parameter $1/k_F a = 1.2$ and measure the condensate fraction. Error bars represent the standard deviation of the bootstrap distribution.

# 4 Conclusion

Our results show that for the production of superfluid quantum gases, simple Fabry-Pérot cavities provide an alternative to high-power running-wave lasers, with all the additional benefits coming along with such optical resonators. By means of a lattice-cancellation scheme, successful evaporation down to the deeply degenerate regime can be carried out inside the cavity trap, preserving the performance in terms of lifetime and heating of the final trap compared to a running-wave optical dipole trap.

Using a resonator with a higher finesse would allow to further reduce the overall laser power needed for evaporation down to powers readily available from standard low-noise lasers, removing the need for fiber amplifiers.

The rigid structure of the cavity trap is enforced by the mirror geometry, which limits the flexibility for future experiments. Nevertheless, the presented technique can be generalized to a cavity driven on several transverse modes at the same time, such that the trap volume could be further varied over a broad range by making use of the intrinsic spatial distribution of the Hermite-Gaussian modes [13]. Furthermore, cavities can be designed to be resonant at several frequencies, as in our case for 532 nm and 1064 nm. Hence it is possible to create repulsive as well as attractive potentials with various spatial structures using different wavelengths and transverse modes. For example, tight confinement can be produced using the $\text{TEM}_{01}$ mode with blue-detuned light, opening the perspective for low-dimensional systems [30].

Last, the cavity-assisted dipole trap is ideally suited for integration with cavity quantum-electrodynamics (cavity QED) experiments where the cavity is used close to atomic resonance for measurements or quantum simulation purposes [31], as employed in our recent work [32]. There it provides inherent relative alignment while minimizing the added complexity to the cavity QED setup that comes with the need for evaporative cooling.

# Acknowledgments

We thank Kevin Roux and Hideki Konishi for contributions to the development of the experimental setup.

**Funding information**   We acknowledge funding from the Swiss State Secretariat for Education, Research and Innovation (Grants No. MB22.00063 and UeM019-5.1). A.F. acknowledges funding from the EPFL Center for Quantum Science and Engineering.

**Data availability**   The data files corresponding to this article are available from the Zenodo repository [33].

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
