# Peer review of "Direct production of fermionic superfluids in a cavity-enhanced optical dipole trap"

_SciPost Physics, doi:SciPost Phys. 18, 133 (2025)_

## Round 1 · Referee Report · Anonymous (Referee 1) · 2025-1-7

Strengths

The paper reports on a novel approach for cooling a Fermi gas into quantum degeneracy by using several longitudinal modes of a high-finess cavity as a dipole trap. This approach is very power-efficient since the cavity recycles the light several thousand times.

Weaknesses

The method described in this paper is very specific and not readily implemented by other experimental setup since it requires a high-finesse cavity inside the vacuum.

Report

The paper is well written and clearly understandable. It makes a very good contribution to advancing the field of ultracold Fermi gases. The results of the paper are sound and well-documented. I recommend publication of the manuscript after minor revisions:

(1) Optical cavities usually exhibit dispersion from the specifics of the coating. As a result, the free spectral range of the cavity is not constant but varies between the different longitudinal modes. For high-finesse cavities this could lead to a diminishing coupling for higher sidebands of the modulation. It would be desirable if the authors could quantify this for their cavity and discuss if this is a limitation to their scheme or not.

(2) I would like to see a brief discussion about the effects of the width of the envelope of the modulation spectrum. Would the use of a mode-locked laser spanning several tens of nm be beneficial or detrimental?

(3) It would be desireable to see the damping coefficient of the dipole oscillations as a function of the modulation parameter.

Recommendation

Publish (easily meets expectations and criteria for this Journal; among top 50%)

  • validity: high
  • significance: good
  • originality: high
  • clarity: top
  • formatting: perfect
  • grammar: perfect

Author:  Tabea Bühler  on 2025-02-11  [id 5207]

(in reply to Report 1 on 2025-01-07)

Dear Referee,

We sincerely appreciate the time you have dedicated to reviewing our manuscript. We would like to thank you for raising insightful points and for your positive evaluation.

Please find below a detailed response to the points you raised. We also indicate the specific parts of the manuscript that have been modified based on your suggestions and remarks, which we believe have improved the quality of the paper.

With best regards,
The Authors

$\textbf{Weaknesses}$
The method described in this paper is very specific and not readily implemented by other experimental setup since it requires a high-finesse cavity inside the vacuum.

Answer: The method is indeed specific to cavity-based systems, however also suitable for cavity-enhanced optical dipole traps that are placed outside vacuum.

$\textbf{Report}$
(1) Optical cavities usually exhibit dispersion from the specifics of the coating. As a result, the free spectral range of the cavity is not constant but varies between the different longitudinal modes. For high-finesse cavities this could lead to a diminishing coupling for higher sidebands of the modulation. It would be desirable if the authors could quantify this for their cavity and discuss if this is a limitation to their scheme or not.

Answer: Indeed, if the properties of the mirror coating change as a function of frequency, this can lead to variations in the free spectral range.

In our case, the frequency range explored in the scope of this scheme is on the order of tens of GHz, translating to less than 0.1 nm. Over this range we did not observe measurable differences in the free spectral range.

In situations where such variations are present, we expect that the proposed scheme will remain effective, although the optimal modulation depth required to suppress the lattice at the center of the cavity is expected to change. The spatial extent of the lattice-free region might be modified as well. However, if the variations in the free spectral range are smaller than the linewidth of the cavity, we neither expect the optimal modulation depth nor the spatial extent of the lattice cancellation to change significantly.

We have added a discussion of this point to the paper text in section 2.1.

(2) I would like to see a brief discussion about the effects of the width of the envelope of the modulation spectrum. Would the use of a mode-locked laser spanning several tens of nm be beneficial or detrimental?

Answer: Using a broad envelope of modulation, for example by using a frequency comb, could indeed be an alternative scheme to successfully cancel the lattice-structure inside the cavity and, potentially, increase the spatial extent over which the lattice structure is suppressed compared to the scheme presented in this paper. This might be particularly desirable if one wishes to cancel the lattice over length scales that start to be comparable to the length of the cavity itself.

However, depending on the free spectral range of the considered cavity and the number of modes involved, such schemes might be affected by variations in the free spectral range, as discussed in the previous point. Additionally, working with a frequency comb would add experimental complexity compared to the method presented in this paper.

While we find the point raised by the referee interesting, we believe that including a discussion on it in the main text would divert focus from the primary message of the manuscript.

(3) It would be desireable to see the damping coefficient of the dipole oscillations as a function of the modulation parameter.

Answer: In cases where we record the dipole oscillations over a large enough time compared to the oscillation period, we do extract parameters such as the oscillation frequency, initial amplitude and damping coefficient by fitting an exponentially damped cosine function to the measured oscillation. This was done for the dataset presented in Fig. 2 b) of the main text, for an optimized modulation depth. From the fit we extract a damping coefficient of 0.09(6) s$^{-1}$, where the duration of the measurement limits the precision with which we can estimate the damping coefficient.

For a varying modulation depth we measure the oscillation amplitude at one point in time, as a proxy for the damping coefficient. This is presented in Fig. 2 a) of the main text. Since we measure the amplitude at one point in time only, we can not perform a fit to extract the damping coefficients in these situations. Nonetheless, we can translate the cloud position at a given time into a damping coefficient using the parameters, such as the initial amplitude, extracted from the fit to the long-lived oscillations at the optimal modulation depth. The extracted damping coefficient $\Gamma$ using the described procedure is plotted in Fig. 1 of the provided reply (please refer to the attached document), together with the atom number and the full width at half maximum of the cloud, as a function of modulation depth. In Fig. 2 of the reply we show a zoomed in version of the damping coefficient as a function of modulation depth.

When converting amplitude into the damping coefficient for modulation depths close to the optimal point, we encounter the problem of the initial amplitude, extracted by the fit to the long-lived oscillations, being smaller than the measured oscillation amplitude at fixed time for a few data points. This leads to a negative damping coefficient and therefore we excluded these data points in Fig. 1 and Fig. 2 of the reply.

For the article, we choose to show the measured cloud position directly, without any mathematical operation or filtering applied to the data. We have changed the formulation in section 2.3 to clarify that we are measuring the oscillation amplitude at a fixed time as a proxy for the damping coefficient, as a function of modulation depth.

Attachment:

figures_reply_r1.pdf

---

## Round 1 · Referee Report · Anonymous (Referee 2) · 2025-1-13

Report

I have reviewed the paper "Direct production of fermionic superfluids in a cavity-enhanced optical dipole trap" by Bühler and collaborators.

I found the paper well written and interesting in its description of cooling techniques to achieve degeneracy conditions for a fermionic system using a cavity enhnaced dipole trap. The technique is well known to be one of the most effective intermediate steps for cooling of lithium fermi gases. This paper addresses improvements on the technique to allow evaporation to degeneracy without the need of extra free space dipole traps, reducing the required infrastructure for such experiments.

I have a number of minor comments to improve the description

-Following Fig.1, it would seem that the frequency of the two odts, ODT1 and 2 is the same, since they are locking on the same cavity mode. This is not really specified in the text and it would probably lead to interference of the two traps, causing instabilities. Is there a particular reason to have them at the same frequency? This point should be clarified.
-In the caption of Fig.1, the term optical potential is used to mean lattice depth or amplitude, without introduction. It would help to define the V symbols in the text.
-Footnotes are given, to specify the models of certain experimental components. The term "exail" is probably referring to the manufacturing company. I would make it more explicit.
-Regarding power stabilization of ODT1 and 2, the authors state that the power is stabilized by monitoring the input and output power of each component. This is not imediately clear. What does the feedback loop stabilize exactly? Moreover, if the two light components are at the same frequency, how are the two differenciated, especially after cavity transmission? Is it with different polarizations? This might be also addressed in a methods section, if too technical.
-Towards the end of Fig.4, in the description of evaporation strategy, the authors say that, while the magnetic field gradient is raised, the optical trap depth is kept constant. I find this unclear. The trap depth needs to decrease for evaporation to take place. I would suggest to indicate that the dipole trap power is kept fixed.
-In the begining of page 6, I found the discussion a bit confusing. First it is reminded to the reader how superfluidity occurs in an attractive Fermi gas (at fixed a), than it is stated that for positive 1/(k_F a) there would be condensation of molecules. Than the actual experimental procedure is described, where evaporation is performed on resonance and a sweep to the repulsive side generates the molecular BEC. It is unclear to the reader what exactly is this procedure. Probably a line break or new paragraph would help here. For example, the sweep to the molecular side is only mentioned at the very end, while it is fundamental to understand the procedure.
-In the caption of FIg.3, a bootstrap distribution is mentioned. This is a well known method to generate error bars, but requires a sampling distribution to be generated. Some indication of how the authors generate the samples would help the reader here.
-In page 7, the discussion about lifetimes states that the they agree with what previously observed by the group in a free space dipole trap situation. I wasn't able to find lifetime measurements in reference 17. A clearer comparison could be done, by computing the expected spontaneous scattering rate for the trap and see if it agrees with the measured lifetime.

The addressing of these comments would improve the paper quality in my opinion.

Recommendation

Publish (meets expectations and criteria for this Journal)

  • validity: good
  • significance: good
  • originality: ok
  • clarity: high
  • formatting: excellent
  • grammar: perfect

Author:  Tabea Bühler  on 2025-02-11  [id 5209]

(in reply to Report 2 on 2025-01-13)

Dear Referee,

We are grateful for the time you invested in reviewing our manuscript and for your positive evaluation. Additionally, we thank you for raising insightful points that have helped to improve the quality of our manuscript and made it more accessible to future readers.

We have carefully addressed the points you raised. Please find below a detailed response to your comments, including a specification of the changes made to the manuscript based on your constructive suggestions.

With best regards,
The Authors

$\textbf{Report}$
-Following Fig.1, it would seem that the frequency of the two odts, ODT1 and 2 is the same, since they are locking on the same cavity mode. This is not really specified in the text and it would probably lead to interference of the two traps, causing instabilities. Is there a particular reason to have them at the same frequency? This point should be clarified.

Answer: We use the same frequency because we use the same laser source for CODT1 and 2. The two could be injected in the cavity on resonance with different longitudinal modes to avoid the aforementioned problem. However, the free spectral range is on the order of several GHz and it is not simple to offset the frequency of one optical dipole trap compared to the other by a multiple of it using acousto-optical modulators. As a consequence, to avoid the described problem, we use orthogonal polarizations in CODT1 and 2. We clarified this in the main text in section 2.1.

-In the caption of Fig.1, the term optical potential is used to mean lattice depth or amplitude, without introduction. It would help to define the V symbols in the text.

Answer: In the new version of the manuscript we introduce the symbol V denoting optical potential depth in the main text.

-Footnotes are given, to specify the models of certain experimental components. The term "exail" is probably referring to the manufacturing company. I would make it more explicit.

Answer: We now made it explicit that we state the part number of the mentioned devices and we separate the manufacturer and part number by a comma to improve clarity.

-Regarding power stabilization of ODT1 and 2, the authors state that the power is stabilized by monitoring the input and output power of each component. This is not imediately clear. What does the feedback loop stabilize exactly? Moreover, if the two light components are at the same frequency, how are the two differenciated, especially after cavity transmission? Is it with different polarizations? This might be also addressed in a methods section, if too technical.

Answer: We measure and stabilize the power injected to the cavity for CODT1 and the power transmitted through the cavity for CODT2. We stabilize the injected power during the loading phase and in the beginning of the optical evaporation, where CODT2 is off. The stabilization of CODT2 takes place during the second evaporation ramp, during which we turn off CODT1.

If desired, the two could be distinguished using polarization-separating optics, as suggested in the referee's comment, allowing them to be operated simultaneously.

We clarified the phrasing in the main text of the manuscript in section 2.1.

-Towards the end of Fig.4, in the description of evaporation strategy, the authors say that, while the magnetic field gradient is raised, the optical trap depth is kept constant. I find this unclear. The trap depth needs to decrease for evaporation to take place. I would suggest to indicate that the dipole trap power is kept fixed.

Answer: We indeed keep the optical dipole trap power constant. We adapted the wording in the main text of the manuscript in section 2.2.

-In the begining of page 6, I found the discussion a bit confusing. First it is reminded to the reader how superfluidity occurs in an attractive Fermi gas (at fixed a), than it is stated that for positive 1/(k_F a) there would be condensation of molecules. Than the actual experimental procedure is described, where evaporation is performed on resonance and a sweep to the repulsive side generates the molecular BEC. It is unclear to the reader what exactly is this procedure. Probably a line break or new paragraph would help here. For example, the sweep to the molecular side is only mentioned at the very end, while it is fundamental to understand the procedure.

Answer: As suggested, we inserted a line break and changed the phrasing in the second paragraph of section 3 to be as clear as possible in the explanation.

-In the caption of FIg.3, a bootstrap distribution is mentioned. This is a well known method to generate error bars, but requires a sampling distribution to be generated. Some indication of how the authors generate the samples would help the reader here.

Answer: Since the signal-to-noise ratio is decreasing as atomic density decreases, we face the problem of a low signal-to-noise ratio on individual pictures for datapoints with a high optical evaporation endpoint in Fig. 3 of the main text. Therefore, we decided to apply the fitting procedure to the average measured atomic densities rather than to individual pictures. To generate the error bars, we obtain a distribution by repeatedly fitting to averaged pictures, where we randomly draw the pictures that are averaged from the dataset.

We added an explanation in section 3 of the manuscript on how we obtain the bootstrap distribution. We also clarified that we are fitting to averaged pictures, motivating the need to perform a bootstrap procedure in order to estimate the error bars.

-In page 7, the discussion about lifetimes states that the they agree with what previously observed by the group in a free space dipole trap situation. I wasn't able to find lifetime measurements in reference 17. A clearer comparison could be done, by computing the expected spontaneous scattering rate for the trap and see if it agrees with the measured lifetime.

Answer: The graph which we were referring to in reference 17 is Fig. 7 b), showing the atom number and the cloud temperature as a function of time. The observed lifetime is on the order of several seconds and a temperature below the critical temperature for superfluidity is measured up to 1 s.

By directly comparing the lifetime and the observed heating of reference 17 with those presented in our manuscript, we assume that the influence of the repeated dispersive probing of the cloud, as it was performed during the measurements of Fig. 7 b) in reference 17, is negligible. This assumption is based on the low photon number per probe scan. A more detailed investigation of the effect of repeated probing is shown in appendix C of reference 17.

In our experiment we never observed dipole trap loss rates that were in agreement with spontaneous scattering rates. We assume that our observed lifetimes are primarily dominated by other mechanisms such as background collisions, three-body recombination processes or intensity noise in the dipole trap independent of the cavity.

The main point in our statement on page 7 of the manuscript is that we observed no change in the time scales associated with atomic losses or heating when transitioning from a free-space optical dipole trap to a cavity-enhanced optical dipole trap.

---

## Round 2 · Referee Report · Anonymous (Referee 2) · 2025-3-6

Report

The points I raised were properly answered, I would suggest publication.
The scope of the paper is a bit narrow, but it's an interesting technical advance in the field.

Recommendation

Publish (meets expectations and criteria for this Journal)

---

## Round 2 · Referee Report · Anonymous (Referee 1) · 2025-3-12

Report

The authors have satisfactorily answered all criticism and they have improved the manuscript. The manuscript can now be published.

Recommendation

Publish (meets expectations and criteria for this Journal)

---

## Round 2 · Author Response

Dear Editor,

We would like to thank the referees for their reports. We have carefully considered the points raised by both referees and have made revisions to the manuscript accordingly. A detailed response addressing each of these points is provided in our replies to their reports.

Below, we summarize the changes that have been implemented, including those made in response to the referees' suggestions. In addition we have performed a recalibration of the magnetic field in the meantime. As a result, we now provide corrected values for the magnetic field and the interaction parameter in Sec. 3 of the manuscript.

We believe that the referees' comments have contributed to improving our manuscript and that it is now ready for publication.

Sincerely,
The Authors

---

## Round 2 · List of Changes

• We added a paragraph in Sec. 2.1 that discusses the stability of the free spectral range of our cavity over the frequency range explored within the scheme presented in this manuscript. The paragraph also discusses potential challenges that may arise in cases where variations of the free spectral range are present.

-We changed the formulation in Sec. 2.3 to clarify that we are measuring oscillation amplitude at a fixed point in time as a proxy for the damping present in the dipole oscillations.

-We added a specification in Sec. 2.1 that CODT1 has orthogonal polarization compared to CODT2.

-We now introduce the symbol V denoting optical potential depth in the main text of the manuscript in Sec. 2.1.

-We changed the format of the footnotes. Stated is the manufacturer of the mentioned device followed by the part number.

-We simplify the phrasing in Sec. 2.1 to clarify that we stabilize the power injected to the cavity for CODT1 and the power transmitted through the cavity for CODT2.

-We change the phrasing in Sec. 2.2 to stress that the optical potential depth is held constant while the magnetic field gradient is raised.

-We added a line break in Sec. 3 to separate the general description of cooling into the superfluid regime from the description of the specific procedure applied in our work. In the second paragraph of Sec. 3 we change the phrasing to increase the clarity of the description of our procedure.

-In Sec. 3 we add a description on how we obtain the bootstrap distribution to generate the error bars. Furthermore we motivate the need for this procedure by stressing that we fit to averaged images.

-In Sec. 3 we add a discussion about the lifetime observed in our experiment and state what we assume are the limiting factors in our case.

-We provide corrected values for the magnetic field and the interaction parameter in Sec. 3 following a recalibration of our magnetic field.

---

## Editorial Decision

published